# FM3VCF: A software library for accelerating the loading of large VCF files in genotype data analyses

Zhen Zuo[1☯], Mingliang Li[1☯], Qi Li[1], Zhuo Li[1], Defu Liu[2], Guanshi Ye[1], You Tang[1,2]*

1 College of Electrical and Information Engineering, Jilin Agricultural Science and Technology University, Jilin, Jilin, China, 2 College of Information Technology, Jilin Agricultural University, Changchun, Jilin, China

☯ These authors contributed equally to this work.
* tangyou9000@163.com

## Abstract

The increasing size of genotype data has led to the loading of VCF files becoming a computational bottleneck in various analyses, including imputation and genome-wide association studies (GWAS). To address this issue, we developed a software library, FM3VCF (fast M3VCF), that utilizes multiple CPU threads to accelerate this process and compress VCF files into the more compact M3VCF format. FM3VCF can convert VCF files into the exclusive data format of MINIMAC4 and M3VCF and can efficiently read and parse data from VCF files. Compared with m3VCFtools, FM3VCF exhibits a speed improvement of approximately 36-fold in the compression of VCF files to the M3VCF format. This acceleration addresses a limitation faced by MINIMAC4 when dealing with datasets containing millions of samples. Furthermore, FM3VCF is approximately 3 times faster than HTSlib, including decompressing and parsing, for reading compressed VCF files. FM3VCF is an effective tool for both compressing VCF files efficiently and accelerating the loading of large VCF files in genotype data analyses. By fully utilizing multiple CPU threads, FM3VCF can significantly reduce the computational burden of various genomic analyses.

## Introduction

Genotype data plays a critical role in modern genomics research, driving advances in understanding complex traits, mapping disease associations, and performing population-based studies [1]. With the expansion of sequencing technologies and the increasing availability of large-scale datasets, researchers face significant computational challenges. Central to these challenges are key analyses such as genome-wide association studies (GWAS) and genotype imputation, which require efficient handling of massive genotype files [2,3]. As the size of datasets grows, the efficiency of data handling, particularly during file loading and parsing, becomes a bottleneck for high-throughput genomic pipelines [4].

**Data availability statement:** The detailed instructions, executable files, source code, and test data simulation code for FM3VCF are freely available, and all raw data required to replicate the study results can be accessed via https://github.com/YOUTLab/FM3VCF. This repository provides all the necessary resources to install, configure, and use FM3VCF for compressing and loading VCF files. The original genetic data used in this study were obtained from the 1000 Genomes Project (https://ftp.1000genomes.ebi.ac.uk/vol1/ftp/release/20130502/).

**Funding:** The Science and Technology Development Plan Project of Jilin Province (20240302074GX). The funders had no role in study design, data collection and analysis, decision to publish, or preparation of the manuscript.

**Competing interests:** The authors have declared that no competing interests exist.

The Variant Call Format (VCF) is the widely adopted standard for storing genotype data due to its flexibility and compatibility across diverse tools and workflows [5]. However, the format's verbosity and the large size of VCF files pose significant limitations, especially in high-dimensional datasets with millions of variants and samples [5]. For example, data parsing and compression are particularly resource-intensive, making existing pipelines less efficient when processing such large-scale datasets [6,7].

Given these limitations, there is a growing need for optimized tools that overcome these computational bottlenecks. Leveraging modern computational advances such as multi-threading can significantly improve the efficiency of genotype data analysis workflows. By accelerating both compression and parsing of large VCF files, researchers can streamline analyses, save computational resources, and focus on the biological insights underlying the data [8,9]. These optimizations are particularly crucial for studies involving datasets with millions of samples, where the computational burden is notably high [10,11].

To address these challenges, we present FM3VCF, a software library designed to overcome the bottlenecks associated with processing large VCF files. FM3VCF offers functionality for converting VCF files into the M3VCF format, a compact format used by MINIMAC4, while providing superior efficiency in file parsing [12]. FM3VCF is specifically optimized to utilize multiple CPU threads, making it capable of handling the demands of large datasets in contemporary genomic research [13].

FM3VCF achieves significant improvements over existing tools. When compressing VCF files to the M3VCF format, FM3VCF is approximately 36 times faster than m3vcftools, enabling rapid file conversion for downstream analyses. Additionally, FM3VCF outperforms HTSlib by threefold in the reading and parsing of compressed VCF files, including decompression tasks [14]. These advancements translate into practical benefits, particularly for studies involving datasets with millions of samples, where the computational burden is notably reduced [15]. By accelerating critical steps in the analysis pipeline, FM3VCF empowers researchers to handle large-scale datasets more effectively, driving progress in genomics research.

## Methods

We aimed to create a high-performance library, FM3VCF, for effectively compressing VCF files to M3VCF files by loading, parsing, and compressing the data. The compression task using m3VCFtools involves three main steps: reading and parsing the VCF file data, compressing and converting the VCF file records to M3VCF file records, and writing the resulting data into the M3VCF file. Since m3VCFtools is single-threaded, all three compression tasks are completed within a single execution workflow.

### Workflow optimization speedup

To improve performance, FM3VCF separates the reading, compressing, and writing processes into distinct stages, allowing each of these stages to be executed in parallel across multiple CPU cores. This workflow optimization enables the three

compression tasks to be processed simultaneously, leading to a faster execution compared to m3VCFtools. For instance, while m3VCFtools requires 9 steps (3 steps for each of the 3 records), FM3VCF can complete the same tasks in only 5 steps (Fig 1), significantly speeding up the process, especially for large datasets.

Let's consider the case where the dataset contains $m$ markers and is divided into $n$ blocks, where each block processes $l$ markers. Thus:

$$n = \frac{m}{l} \tag{1}$$

The time required for reading, compressing, and writing each block are denoted as $T_{read}$, $T_{compress}$ and $T_{write}$, respectively.

In the original sequential workflow (i.e., before optimization), the total time is:

$$T_{old} = n \cdot (T_{read} + T_{compress} + T_{write}) \tag{2}$$

With the optimized workflow, where reading, compressing, and writing are handled in parallel across the $n$ blocks, the total time is:

$$T_{new} = 1 \cdot T_{read} + n \cdot T_{compress} + 1 \cdot T_{write} \tag{3}$$

This results in a significant reduction in time spent on reading and writing, which are now only handled once for the entire dataset, compared to the previous $n$ times in the original workflow.

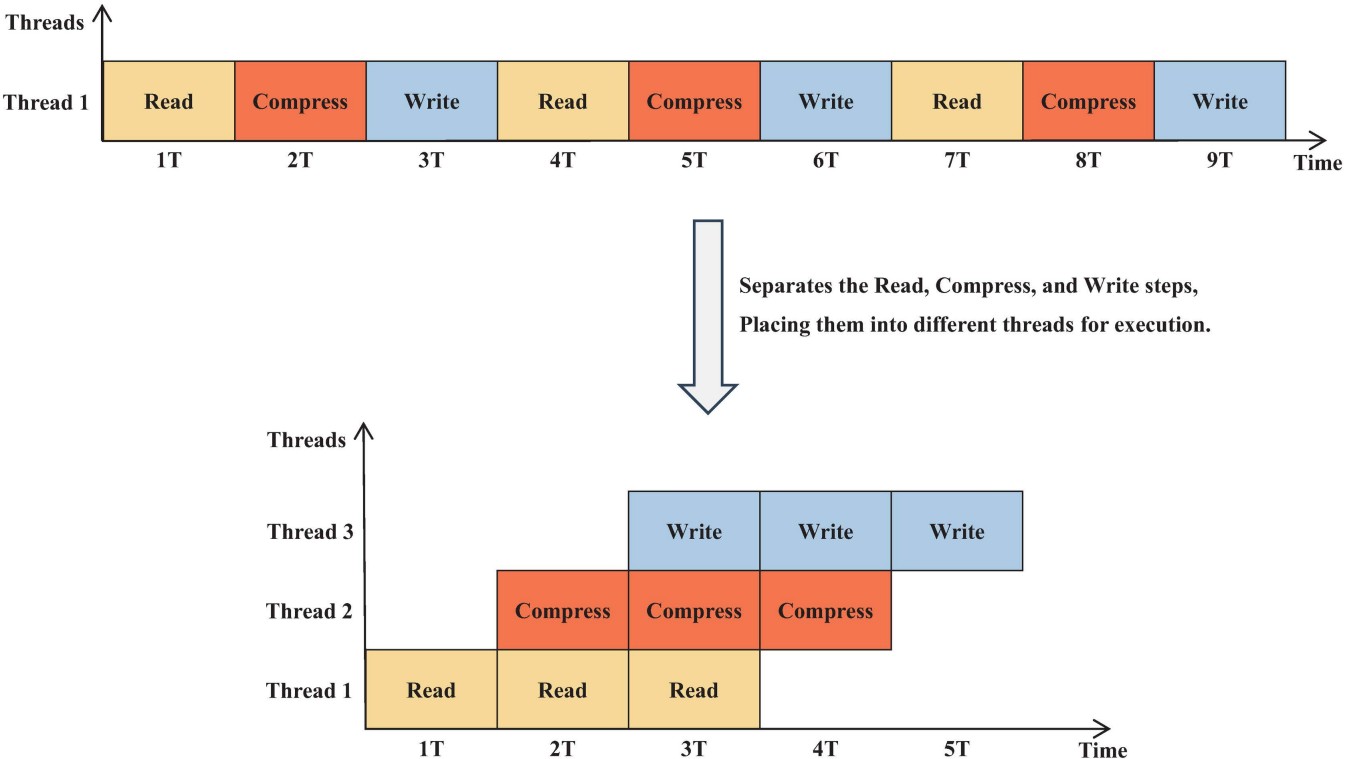

**Fig 1. Transition from Single-Threaded to Multi-Stage Workflow Execution.**

The time savings can be quantified as:

$$T_1 = (n-1) \cdot (T_{read} + T_{write}) \qquad (4)$$

Thus, the optimized workflow reduces the time taken by removing redundant read and write operations for each block, leading to improved efficiency, particularly for large datasets.

**Parallel compression using pthreads**

The Compress step is the most time-consuming among the three processes. To address this, a multi-threaded approach using the pthread library [16] is employed to execute the Compress step in parallel across multiple CPU cores (Fig 2a). This parallelization results in a more balanced distribution of processing time across the Read, Compress, and Write steps. By distributing the compressing task across multiple threads, the system can leverage the computational power of multiple CPU cores, significantly reducing the overall time spent in the compressing phase.

**a**

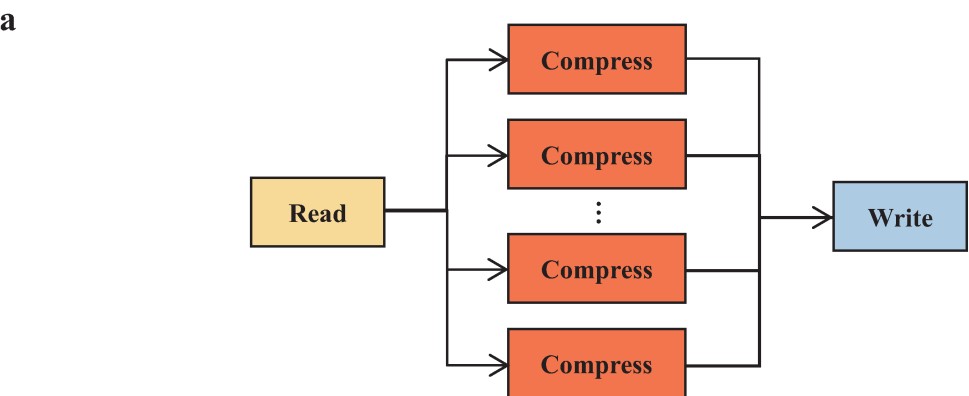

**b**

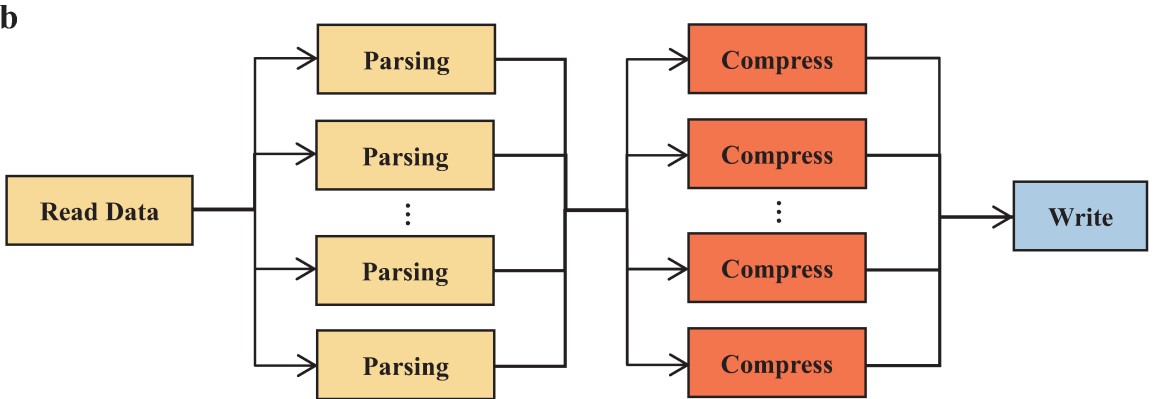

**Fig 2. Optimizing Data Compression and Reading Efficiency through Parallel Processing Strategies.** (a) Parallel Compression Using pthreads. (b) Optimized Reading with Parallel Parsing.

Let the number of threads used in the Compress step be $p$. In the original, single-threaded approach, the total time for the Compress step is $T_{compress}$, which is repeated for each block of data. With the parallelized Compress step, assuming an ideal scenario (perfect parallelization with no overhead), the total time for compressing is reduced to:

$$T_{compress}' = \frac{T_{compress}}{p} \qquad (5)$$

The time savings by parallelizing the Compress step can be quantified as:

$$T_2 = \frac{(p-1) \cdot T_{compress}}{p} \qquad (6)$$

This speedup shows that, with $p$ threads, the compression time can be reduced by a factor of $\frac{p-1}{p}$, leading to a significant improvement in the overall compression process.

**Optimized reading with parallel parsing**

The Read step is divided into two phases: Reading Data and Parsing. Parsing consumes the majority of the time in this step. To accelerate this process, the OpenMP library [17] is used to run the Parsing phase in parallel across multiple threads (Fig 2b). This results in a significant increase in processing speed for the Read step, especially when dealing with large datasets, as the parallelization of the Parsing phase speeds up the overall data loading and preparation process.

Let the number of threads used for Parsing be $q$. In the original, single-threaded approach, the total time for Parsing is $T_{parse}$, and the time for Reading Data is $T_{load}$. When the Parsing phase is parallelized, the total time for the Reading step becomes:

$$T_{read}' = T_{load} + \frac{T_{parse}}{q} \qquad (7)$$

Thus, The time savings by parallelizing the Parsing step is:

$$T_3 = \frac{(q-1) \cdot T_{parse}}{q} \qquad (8)$$

This shows that, by parallelizing the Parsing phase, the total time for the Read step is reduced, with the amount of reduction depending on the number of threads $q$ used. As the number of threads increases, the time spent on Parsing decreases, leading to faster overall reading and parsing performance.

In the comparison between zm3vcf and M3VCF, the total time savings, denoted as:

$$T_s = T_1 + T_2 + T_3 \qquad (9)$$

After workflow optimization and parallelization of the compression and parsing steps, the overall complexity of the process can be analyzed by considering the individual components involved: reading, compression, and writing. Each step has its own complexity: $O_{read}$ for reading, $O_{compress}$ for compression, and $O_{write}$ for writing. Initially, the overall complexity before optimization is expressed as:

$$O_{total} = O_{read} + O_{compress} + O_{write} \qquad (10)$$

With workflow optimization and the introduction of multi-threading for both the compression (using $p$ threads) and parsing (using $q$ threads) steps, the overall complexity is improved. The complexity of reading and parsing is reduced due to

the parallel execution, while compression benefits from multi-threading. Thus, the overall complexity after optimization becomes:

$$O_{total}' = \frac{l}{m} \cdot \left[ O_{read} + \frac{1}{q} \cdot O_{prase} + O_{write} \right] + \frac{1}{p} \cdot O_{compress}$$

(11)

Here, $m$ represents the total number of markers, $l$ represents the number of markers processed in each block, $p$ is the number of threads used for compression, and $q$ is the number of threads used for parsing.

When focusing on the improvement in the compression step, the overall complexity can be simplified to:

$$O = \frac{1}{p} \cdot O_{compress}$$

(12)

This simplification highlights the primary advantage of multi-threading in the compression step, where the time complexity is significantly reduced by distributing the task across multiple threads. This leads to an overall faster execution, especially as the dataset size increases.

When multi-threading is not enabled, the primary time savings come from $T_1$, as the optimization in reading and parsing reduces redundant operations and accelerates the data loading phase. In this scenario, the compression step remains single-threaded, meaning the time savings from $T_2$ are not significant. However, when multi-threading is enabled, the compression step, which is the most time-consuming process, benefits the most from parallel execution. As a result, the savings in $T_2$ become the dominant factor, significantly improving the overall performance, especially as the dataset size increases. This is because the larger the dataset, the higher the acceleration factor achieved through parallel compression. Therefore, the time savings from $T_2$ are more pronounced in multi-threaded execution, making the acceleration effect increasingly evident with larger data volumes. Optimized Reading with Parallel Parsing plays a time-saving role in all reading processes, especially when dealing with larger datasets. The time saved in the optimized writing process, $T_3$, effectively reduces the overall runtime, thereby improving the efficiency of the entire data processing pipeline.

## Results

To illustrate the performance of M3VCF compression and VCF loading, we present examples of command line and library interface usage. It is important to note that M3VCF only accepts bi-allelic SNPs for compression. Detailed instructions, executable files, and source code can be accessed at https://github.com/YOUTLab/FM3VCF.

VCF Reading and M3VCF Compression: To load a large number of genotype records from a VCF file using the library interface, the following steps are performed (Box 1): First, the VCF format structure is initialized (lines 4–6), followed by opening the VCF file (lines 7–9) and reading the header (line 10). There are two optional functions available for reading the variant records either variant by variant (line 11) or block by block (line 13), which allows efficient loading of large datasets. Finally, the memory is released (line 18 or line 19), and the VCF file is closed (line 20). To specify the VCF file format, the parameter "FILE_MODE_GZ" is used for compressed files with the ".vcf.gz" suffix, while "FILE_MODE_NOR-MAL" is used for uncompressed files with the ".vcf" suffix in the vcfFileOpen() function. The VCF reading interface parses only the genotype (GT) and dosage (DS) information in the FORMAT field. Users can choose to load both by specifying the parameter "P_DS|P_GT" or load only GT by using "P_GT" in the vcfFileOpen() function. In the block reading function vcfFileReadDataBlock(), the number of variants in each block can be specified by the parameter "int numLines," which enables the utilization of multiple threads for parsing.

After loading data from the VCF file, compression can be implemented using the following examples in Box 2. The first two lines demonstrate the command line usage for compression (line 1) and decompression (line 2). The "-o" flag specifies the output file name, "compress" refers to M3VCF file compression, the "-m/M" flag specifies whether the output file is

compressed or not, and "convert" is used for decompressing M3VCF files to VCF format. For the C/C++ library interface usage, lines 3–28 provide the necessary steps. The initial steps involve defining the block size (number of variants; line 4) and specifying the input and output file paths (lines 5–7). Then, the variable configuration for compression (lines 9–16) and decompression (lines 20–24) should be specified before launching the compression (lines 17–19) and decompression (lines 25–27) functions. After implementation, FM3VCF library automatically releases the allocated memory. For both compression and decompression variable configurations, the VCF (lines 11, 24) and M3VCF file (lines 13, 22) compression statuses should be specified, controlled by "FILE_MODE_NORMAL" and "FILE_MODE_GZ". The compression function supports multiple threads, and users can specify the block size, number of threads, and memory size (lines 14–16) where the value 0 represents the optimized default setting (lines 15–16). Instructions for code compilation for Boxs 1–2 can be found at https://github.com/YOUTLab/FM3VCF/blob/main/FM3VCF_instruction.pdf.

---

Box 1. VCF reading and parsing (library interface).

```
1    #include "vcflib.h"

2    const char *file_path = "ALL.chr22.20Markers.10Samples.vcf.gz";

3    int main(int argc,char** argv){

4    VCF_FILE fp_read;

5    DATA_LINE dataLine={0};

6    DATA_BLOCK dataBlock={0};

7    if(VCF_ERROR==vcfFileOpen(&fp_read,file_path,FILE_MODE_GZ,P_DS|P_GT)){

8    printf("the vcf file:%s, open error\n", file_path);

9    return 1;}

10   vcfFileReadHead(&fp_read);

11   vcfFileReadDataLine(&fp_read,&dataLine);

12   printDataLine(&dataLine);

13   vcfFileReadDataBlock(&fp_read,&dataBlock,10);

14   printf("the first line of data block:\n");

15   printDataLine(&(dataBlock.dataLines[0]));

16   printf("the last line of data block:\n");

17   printDataLine(&(dataBlock.dataLines[dataBlock.numDataLines-1]));

18   clearDataLine(&dataLine);

19   clearDataBlock(&dataBlock);

20   vcfFileClose(&fp_read);

21   return 0;

}
```

Box 2. M3VCF compression (command line and library interface).

```
1   ./zM3vcf compress testFile/ALL.chr22.20Markers.10Samples.vcf.gz -O m -o testFile/ALL.chr22.20Markers.10Sam-
    ples.m3vcf.gz

2   ./zM3vcf convert testFile/ALL.chr22.20Markers.10Samples.m3vcf.gz -O m -o testFile/ALL.chr22.20Markers.10Sam-
    ples.CL.vcf.gz

3   #include "m3vcf/m3vcf.h"

4   int default_block_size = 10;

5   const char *vcf_file_path = "testFile/ALL.chr22.20Markers.10Samples.vcf.gz";

6   const char *m3vcf_flie_path = "testFile/ALL.chr22.20Markers.10Samples.m3vcf.gz";

7   const char *if_vcf_file_path = "testFile/ALL.chr22.20Markers.10Samples.IF,vcf.gz";

8   int main(int argc,char** argv){

9   M3VCF_COMPRESS_ARGS com_args;

10  com_args.vcfFileName = vcf_file_path;

11  com_args.vcfFileType = FILE_MODE_GZ;

12  com_args.m3vcfFileName = m3vcf_flie_path;

13  com_args.m3vcfFileType = FILE_MODE_GZ;

14  com_args.bufferSize = default_block_size;

15  com_args.thread_num = 0;

16  com_args.memory_limit = 0;

17  if(M3VCF_OK! = vcfCompressToM3vcf(&com_args)){

18  fprintf(stderr,"vcffile:[%s] compressed error!\n",com_args.vcfFileName);

19  return 1;}

20  M3VCF_CONVERT_ARGS con_args;

21  con_args.m3vcfFileName = m3vcf_flie_path;

22  con_args.m3vcfFileType = FILE_MODE_GZ;

23  con_args.vcfFileName = if_vcf_file_path;

24  con_args.vcfFileType = FILE_MODE_GZ;

25  if(M3VCF_OK! = m3vcfConvertToVcf(&con_args)){

26  fprintf(stderr,"m3vcffile:[%s] convert error!\n",con_args.m3vcfFileName);

27  return 1;}

28  return 0;}
```

To evaluate the performance of FM3VCF, we used the chromosome 1 and chromosome 2 files from the 1000 Genomes Project [18] (ftp://ftp.1000genomes.ebi.ac.uk/vol1/ftp/release/20130502/). Using ped-sim [19], the program to simulate pedigree structures, we simulated five datasets with 550, 2,750, 5,500, 11,000, and 22,000 samples based on genetic rules. Each dataset contains 13,435,471 variants. To further assess FM3VCF's performance across different sample sizes, the simulated datasets were scaled according to 1x, 5x, 10x, 20x, and 40x sample sizes. These simulated datasets allow for a comprehensive evaluation of FM3VCF's compression and loading performance when handling genomic data at various scales.

For the compression of M3VCF files from compressed VCF files, we employed different numbers of threads (6, 8, 10, 20, and 30) to conduct the tests. For the dataset with 22,000 samples, m3VCFtools took 570.9 hours for compression, while FM3VCF completed the task in 15.5 hours using 30 threads. Therefore, FM3VCF demonstrated approximately 36.8 times faster performance compared to m3VCFtools (Fig 3a). Under the condition that hardware resources allow, as the number of threads increases, the computation time gradually decreases (Fig 3b), and the more threads there were, the more memory was required (Fig 3c).

We also evaluated the performance of VCF file loading using the same datasets. When employing the block loading method, FM3VCF took 269.3 minutes, which was approximately 3 times faster than HTSlib with 13,435,471 samples and 10 threads, specifically for loading data and parsing GT only (Fig 3d). Additionally, we further tested random data containing both GT and DS, and the time consumption for parsing both GT and DS was almost the same as when only parsing GT (S1 Fig d).

## Discussion

The experimental results show that FM3VCF, using 30 threads, demonstrated approximately 36.8 times faster performance compared to m3VCFtools (Fig 3a). Theoretically, if only multi-threading were utilized, the maximum speedup would be limited to less than 30 times. However, due to the Workflow Optimization Speedup, a significant amount of read and write time was saved, as illustrated by Equation 4. This optimization allowed for a performance improvement beyond the theoretical limit of multi-threading alone.

The observed phenomenon in Fig 3b, where the speedup trend flattens between 20 and 30 threads, can be explained by Amdahl's Law [20]. This law describes the theoretical limit of parallelization efficiency, highlighting that some portions of a task cannot be parallelized, which limits the overall performance gain. In the case shown in the supporting materials (S1 Fig b), the performance difference between 20 and 30 threads is almost negligible, which aligns with the concept of saturation as the number of threads increases.

Amdahl's Law gives the theoretical speedup $S_{Amdahl}$ as:

$$S_{Amdahl} = \frac{1}{(1-\alpha)+\frac{\alpha}{p}}$$

(13)

Where: $\alpha$ is the fraction of the task that can be parallelized. $p$ is the number of threads.

The phenomenon of saturation at larger thread counts is consistent with the predictions of Amdahl's Law, where non-parallelizable parts of the task (such as I/O operations and possible thread management overhead) limit the performance improvement. Although FM3VCF demonstrates significant performance improvements with multi-threading, especially when handling large datasets, hardware and task-specific limitations cause thread acceleration to reach saturation more easily when dealing with larger sample sizes, resulting in diminishing returns from increasing the number of threads.

The process of reading, compressing, and writing data is a pipeline. If any part of the pipeline experiences a bottleneck, the entire flow will be halted, causing the system to wait. The use of multi-threading in the compression step is aimed at ensuring that the speed of each stage in the pipeline—reading, compressing, and writing—remains consistent, preventing any waiting or idle time. By parallelizing the compression process, the system can better match the throughput of each

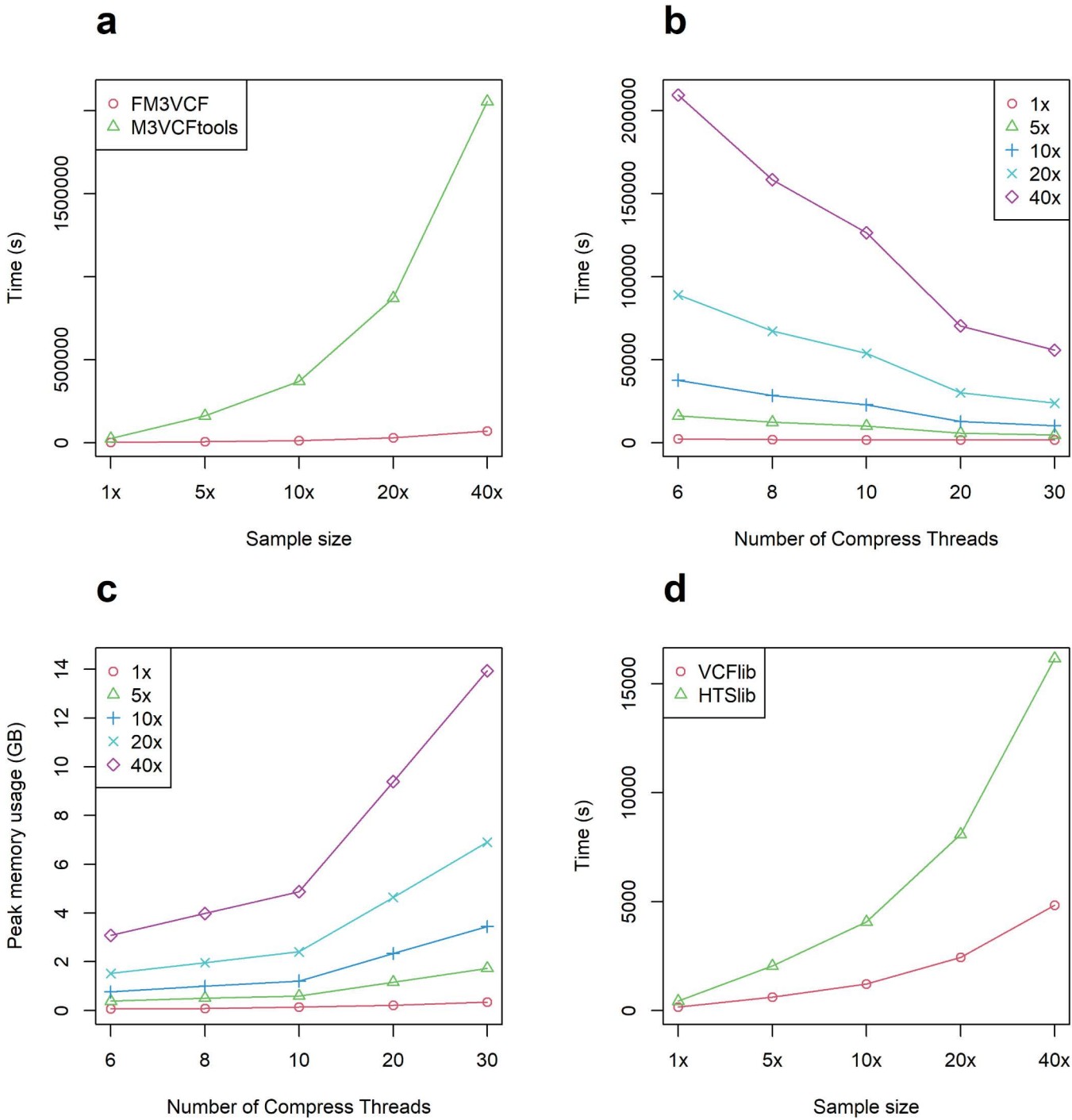

**Fig 3. Performance evaluation of FM3VCF.** (a) The compression time for VCF files of different sizes is depicted in the figure. The green line represents the results obtained using m3VCFtools, while the red line represents the results obtained using FM3VCF. (b) The multithread computation time of FM3VCF. The results for sample sizes of 40x, 20x, 10x, 5x, and 1x are represented by the purple, cyan, blue, green, and red lines, respectively. (c) The multithread peak memory usage by FM3VCF. The results for sample sizes of 40x, 20x, 10x, 5x, and 1x, denoted by the purple, cyan, blue, green, and red lines, respectively. (d) Reading and parsing times for VCF files of different sizes. The green line corresponds to the reading time for different sample sizes using HTSlib. The red line shows the time needed for parsing GT by FM3VCF.

stage, thus eliminating delays and optimizing the overall pipeline performance. This approach ensures that the entire workflow operates at maximum efficiency, preventing any stage from becoming a limiting factor.

In addition to thread utilization, the choice of compression parameters, particularly block size, plays a crucial role in the performance of FM3VCF. By adjusting the block size, both data processing efficiency and the effective use of multi-threading are influenced. Smaller blocks may lead to more frequent thread synchronization and higher overhead, while larger blocks can reduce the number of synchronization points. However, if the data is unevenly distributed, it may result in load imbalance. Therefore, optimizing the block size is essential to finding the best balance between parallelization efficiency and I/O performance. For specific files, if the number of markers is large and the sample size is small, it may be beneficial to increase the default_block_size parameter to achieve a more balanced load distribution.

## Conclusion

We have developed FM3VCF, a powerful library that allows efficient loading of data from VCF files and compression into the M3VCF format. This software provides both a command line interface and a C/C++ library interface, offering researchers the flexibility and effectiveness needed to conduct genomic analysis using large datasets. With FM3VCF, researchers can effectively handle and analyze genomic data, enabling them to delve into complex genetic studies and derive valuable insights.

## Supporting information

**S1 Fig. Performance evaluation of FM3VCF on a randomly simulated dataset.** The dataset is derived from chromosome 22 of the 1000 Genomes Project, containing 1,092 samples and 494,328 variants (http://ftp.1000genomes.ebi.ac.uk/vol1/ftp/release/20110521/ALL.chr22.phase1_release_v3.20101123.snps_indels_svs.genotypes.vcf.gz). (a) The figure shows the compression time for VCF files of different sizes. The blue line represents the results obtained using m3VCFtools, while the red line represents the results obtained using FM3VCF. (b) Multi-thread computation time for FM3VCF. The results for sample sizes of 80x, 40x, 20x, 10x, 5x, and 1x are represented by the purple, cyan, blue, green, red, and orange lines, respectively. (c) Multi-thread peak memory usage by FM3VCF. The results for sample sizes of 80x, 40x, 20x, 10x, 5x, and 1x are represented by the purple, cyan, blue, green, red, and orange lines, respectively. (d) Reading and parsing times for VCF files of different sizes. The green line corresponds to the reading time for different sample sizes using HTSlib. The red line represents the time required for parsing GT and DS by FM3VCF, while the blue line shows the time needed for parsing GT by FM3VCF.
(PDF)

## Acknowledgments

We would like to express our sincere gratitude to Meng Huang for his expert guidance and invaluable advice throughout the research and writing process. We also extend our thanks to the peer reviewers whose suggestions have greatly improved the quality of this manuscript.

## Author contributions

**Conceptualization:** You Tang.

**Data curation:** Zhen Zuo, Mingliang Li, Zhuo Li.

**Funding acquisition:** Guanshi Ye, You Tang.

**Methodology:** Zhen Zuo, Mingliang Li, You Tang.

**Project administration:** You Tang.

**Software:** Zhen Zuo, Mingliang Li, Defu Liu.

**Supervision:** Guanshi Ye, You Tang.

**Validation:** Qi Li.

**Writing – original draft:** Mingliang Li.

**Writing – review & editing:** Qi Li, You Tang.

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
