## [Decision Letter · Decision Letter 0]

18 Nov 2024

Dear Dr. Tang,

Thank you for submitting your manuscript to PLOS ONE. After careful consideration, we feel that it has merit but does not fully meet PLOS ONE’s publication criteria as it currently stands. Therefore, we invite you to submit a revised version of the manuscript that addresses the points raised during the review process.

**The reviewers’ comments pointed out that a major revision is required to improve the current version of the manuscript. One of the most critical concerns regards the literature review. The authors should also discuss the novelty and main contribution of the proposed method, as well as its limitations. Please refer to the reviewers’ reports for detailed comments, which could help improve the current version of the manuscript. Please carefully address (and reply to) all the comments raised by all reviewers (this is mandatory).**

We look forward to receiving your revised manuscript.

Kind regards,

Andrea Tangherloni

Academic Editor

PLOS ONE

**Journal Requirements:**

The Science and Technology Development Plan Project of Jilin Province (20240302074GX)

**Additional Editor Comments:**

The reviewers’ comments pointed out that a major revision is required to improve the current version of the manuscript. One of the most critical concerns regards the literature review. The authors should also discuss the novelty and main contribution of the proposed method, as well as its limitations. Please refer to the reviewers’ reports for detailed comments, which could help improve the current version of the manuscript. Please carefully address (and reply to) all the comments raised by all reviewers (this is mandatory).

Reviewers' comments:

Reviewer's Responses to Questions

**Comments to the Author**

1. Is the manuscript technically sound, and do the data support the conclusions?

Reviewer #1: No

Reviewer #2: Yes

Reviewer #3: Partly

2. Has the statistical analysis been performed appropriately and rigorously?

Reviewer #1: No

Reviewer #2: Yes

Reviewer #3: No

3. Have the authors made all data underlying the findings in their manuscript fully available?

Reviewer #1: No

Reviewer #2: Yes

Reviewer #3: Yes

4. Is the manuscript presented in an intelligible fashion and written in standard English?

Reviewer #1: No

Reviewer #2: Yes

Reviewer #3: No

**Reviewer #1:**  The research work lacks sufficient explanation and validation, with the background study falling short of standards. The authors have reviewed a minimal number of papers, resulting in a weak background. Additionally, the discussion and validation of the results are inadequate. The paper fails to address the significance, novelty, and contribution of the study, leading to its rejection. The authors neglected to discuss these critical components, which are essential for the research's credibility and impact.

**Reviewer #2:**  The authors present a novel methodology and contribution with FM3VCF by utilising multi-threading to efficiently read large VCF files and also compresses them into the M3VCF format. The findings show promise as the proposed method is significantly faster than other widely used methods for VCF compression.

Strong Aspects:

1. Research topic is novel with a potential impact in the field.

2. Paper writing style, with reference to technical English, is quite good.

3. Robust methodology with rigorous testing.

The work prima facie is of good quality; however, the following minor negatives need to be corrected:

1. Not enough literature review/related work. While the methods are incredibly detailed, a more comprehensive literature review and a review of the current state-of-the-art would make the manuscript a better read.

**Reviewer #3: ** Dear authors

I was asked to prepare a review of your paper.

In my humble opinion this is a very technical paper that refers to important from practical point of view problems, but on the other hand it lacks of scientific content.

Below there are some of my suggestions, remarks and comments:

1. In Results paragraph (line 22) you wrote:

FM3VCF exhibits a speed improvement of approximately 20-fold in the compression of VCF files to the M3VCF format.

This result was obtained basing on set of experiments, but it is not justified from the analytical analysis of your proposed method. Such analysis should be based on:

(i) workflow of proposed method - you only gave a description expressed in words (lines 88-101) with reference to program code

(ii) complexity analysis of proposed algorithm/approaches

(iii) limitations that can appear in reference to input set, hardware used for test,

Your simulations are not final proofs of obtained results, but they confirm from practical side, what was achieved. Please expand this including fore theoretical framework.

2. Line 26, there is:

FM3VCF is a powerful tool for both efficiently compressing VCF files and accelerating the loading of large VCF files in genotype data analyse.

I would prefer not to use such words as 'powerful'; we were able to see the accelaration effects but without analytical proofs that your approach will work for all input data (input cases) such strong statements are not acceptable. I'm not saying that your is wrong or does not work, but this cannot be based only on simulations - see also remark 6.

3. Technical remarks -> in introduction please separate references from words and put space between the words and literature references according to: xxx[1] -> xxx [1].

4. Lines 150-151, we have:

This software provides both a command line interface and a C/C++ library interface,

I cannot see this in paper.

5. Is it possible to express the resuts of compressing files with the use of compressing threads in terms of mathematical formula. From Fig. 3 b) we can see after the quick increase of characteristics there is no difference bewteen 20 and 30 threads. What is the reason of such behavior? Line 131 gives information about 'saturation'. Please explain and expand this

6. Important remark !!! (line 123)

We have: Five datasets were created by duplicating the original dataset with 5-, 10-, 20-, 40-, and 80-fold duplicated samples

If my way of understanding is correct, you duplicated several (teens) times exactly the same input set. For such set it is very easy to expect that when it will be compressed the result of compression will be significant and impressive. But this comes from simple properties of theory of information. How your approach will work when such approach won't be used, or the coppied parts will be randomly shuffled.

7. What will be the influence of different compression parameters on your approach performance? I see from the listing that you used:

default_block_size = 10;

com_args.bufferSize = default_block_size;

**Do you want your identity to be public for this peer review?** For information about this choice, including consent withdrawal, please see our Privacy Policy

Reviewer #1: No

Reviewer #2: No

Reviewer #3: No

---

## [Author Response · Author response to Decision Letter 1]

7 Mar 2025

Dear Dr. Tangherloni and Reviewers,

We sincerely thank you for the time and effort you have dedicated to reviewing our manuscript titled "FM3VCF: A Software Library for Accelerating the Loading of Large VCF Files in Genotype Data Analyses"** (Manuscript ID: PONE-D-24-38236). Your constructive feedback has been invaluable in enhancing the quality and clarity of our work. Below, we provide detailed responses to each of the comments raised by the Academic Editor and the reviewers.

We would like to thank the academic editor and reviewers for recognizing the value of this manuscript. In response, we have made revisions to address the points raised:

1. We have added a more comprehensive review of the related research, covering the background of the study, recent advancements in VCF file processing, and genotype data analysis tools.

2. We have provided a detailed explanation of the unique aspects of FM3VCF, including an introduction to the workflow and theoretical framework, clearly highlighting its novel features and major contributions to the field.

3. We have included additional experiments using more diverse datasets to further validate the effectiveness of our tool.

4. We have conducted a more detailed discussion, analyzing and summarizing the impact of compression parameters such as the number of threads, block size, and file size on the performance of the tool, providing a more comprehensive perspective and guidance for future research.

These improvements aim to bring the manuscript into closer alignment with PLOS ONE’s publication standards.

Responses to Reviewer #1

Reviewer #1 Overall Feedback: The research work lacks sufficient explanation and validation, with the background study falling short of standards. The authors have reviewed a minimal number of papers, resulting in a weak background. Additionally, the discussion and validation of the results are inadequate. The paper fails to address the significance, novelty, and contribution of the study, leading to its rejection. The authors neglected to discuss these critical components, which are essential for the research's credibility and impact.

Comment 1:

The research work lacks sufficient explanation and validation, with the background study falling short of standards. The authors have reviewed a minimal number of papers, resulting in a weak background.

Response:

Thank you for the valuable feedback from the reviewer. We have made significant revisions to strengthen the manuscript.

Firstly, we have expanded the literature review: We have significantly broadened the scope of the literature review to include a more comprehensive discussion of relevant research. This now covers:

The importance and role of genotype data in genomics research, particularly in studies such as genome-wide association studies (GWAS) and genotype imputation (lines 33-35).

The computational challenges researchers face as dataset sizes grow, particularly regarding VCF file handling, loading, and parsing (lines 35-40).

A detailed review of the limitations of the Variant Call Format (VCF) in handling large datasets with millions of variants and samples, and the challenges of data parsing and compression (lines 41-46).

Secondly, we emphasized the growing need for optimized tools: We highlighted the increasing demand for optimized tools that can handle large-scale genomic data efficiently. The background now includes discussions on how multi-threading and modern computational advances can address these challenges and improve the efficiency of genotype data analysis workflows (lines 47-53).

Lastly, we provided a more detailed introduction to FM3VCF: We have further clarified the specific contributions and innovations of FM3VCF. The manuscript now clearly presents FM3VCF as a tool designed to overcome the bottlenecks of processing large VCF files by utilizing multi-threading for improved efficiency in both compression and parsing, making it highly suited for large datasets (lines 54-59).

These revisions provide a more robust context for our research, validate the need for the tool, and demonstrate how FM3VCF addresses existing limitations. We believe that these improvements significantly strengthen the manuscript and ensure a more comprehensive background.

Comment 2:

The discussion and validation of the results are inadequate.

Response:

We have added more experiments using more appropriate and diverse datasets to further validate the effectiveness of FM3VCF (lines 210-218). Additionally, we conducted further tests to evaluate FM3VCF’s performance across different datasets and scenarios (lines 219-240, lines 289-303). Moreover, we expanded the discussion section to provide a deeper analysis of how parameters such as the number of threads, block size, and file size affect FM3VCF's performance. We explained, using Amdahl’s Law, how performance gains tend to diminish as the number of threads increases, and how hardware and task-specific limitations cause the acceleration effect to reach saturation more easily (lines 242-271). We also analyzed the critical role of optimizing block size to balance parallelization efficiency and I/O performance. A well-chosen block_size parameter is key to achieving a more balanced load distribution (lines 272-280).

These additions provide a clearer interpretation of our results in the context of existing methods and highlight the practical implications and potential of our findings.

Comment 3:

The paper fails to address the significance, novelty, and contribution of the study.

Response:

We appreciate the valuable feedback provided by the reviewer. In response to the comment regarding the insufficient explanation of the significance, novelty, and contribution of the study, we have made the following revisions:

Significance of the Study: In the introduction, we further clarified the background and significance of FM3VCF, emphasizing the importance of genotype data in modern genomics research, particularly in key analyses such as genome-wide association studies (GWAS) and genotype imputation (lines 33-40). Additionally, we discussed how, as dataset sizes grow, VCF file processing has become a major computational bottleneck in high-throughput genomics (lines 41-46), highlighting the need for the development of efficient tools (lines 47-53).

Novelty: We explicitly pointed out the novelty of FM3VCF, particularly the breakthroughs in file compression and loading efficiency. By optimizing the workflow (lines 70-100) and utilizing multi-threading technology (lines 101-157), FM3VCF achieves significantly faster compression speeds compared to existing tools like m3VCFtools (lines 219-240). Moreover, through improvements in block size adjustments and parallel parsing methods, FM3VCF provides a more efficient data processing solution(lines 241-280). We highlighted how these innovations effectively address the limitations of current methods when dealing with large-scale genomic data.

Contribution: We further emphasized FM3VCF's contribution to genomic data analysis, especially in enhancing processing efficiency for VCF files containing large numbers of variants and samples. By incorporating multi-threaded parallel processing and optimized file format conversion, FM3VCF not only accelerates data loading and compression but also reduces computational resource consumption, thus advancing large-scale genomic research (lines 54-67).

Through these revisions, we have provided a clearer presentation of the significance, novelty, and contribution of FM3VCF, helping readers better understand how our work fills the gap in existing tools and contributes to genomic analysis.

We hope these revisions meet your expectations and look forward to your further feedback.

Responses to Reviewer #2

Comment:

The authors present a novel methodology and contribution with FM3VCF by utilising multi-threading to efficiently read large VCF files and also compresses them into the M3VCF format. The findings show promise as the proposed method is significantly faster than other widely used methods for VCF compression.

Strong Aspects:

1. Research topic is novel with a potential impact in the field.

2. Paper writing style, with reference to technical English, is quite good.

3. Robust methodology with rigorous testing.

The work prima facie is of good quality; however, the following minor negatives need to be corrected:

1. Not enough literature review/related work. While the methods are incredibly detailed, a more comprehensive literature review and a review of the current state-of-the-art would make the manuscript a better read.

Response:

Thank you for your valuable feedback, particularly regarding the literature review section. We appreciate your suggestion to make the manuscript more comprehensive by providing a more detailed and thorough review of the related work. In response, we have made significant revisions to the manuscript to address this concern.

Firstly, we have expanded the literature review to include a more in-depth discussion of relevant research. We now emphasize the importance of genotype data in genomics, particularly in studies such as genome-wide association studies (GWAS) and genotype imputation (lines 33-40). Additionally, we extended the review of existing tools and methods for VCF file processing and genotype data analysis, highlighting the challenges and solutions related to handling large-scale datasets (lines 41-53). Furthermore, we discussed the limitations of current methods and tools to provide a solid foundation for FM3VCF’s innovation and contributions (lines 53-67).

By expanding the literature review, we have provided a more robust theoretical context for our research and clearly demonstrated FM3VCF’s unique value and contribution to overcoming the limitations of existing tools.

We believe these revisions significantly strengthen the manuscript and enhance its readability, making the background and methodology sections more coherent and comprehensive. Thank you again for your insightful comments, which have greatly improved the manuscript.

Responses to Reviewer #3

Comment 1:

In Results paragraph (line 22) you wrote:

FM3VCF exhibits a speed improvement of approximately 20-fold in the compression of VCF files to the M3VCF format.

This result was obtained basing on set of experiments, but it is not justified from the analytical analysis of your proposed method. Such analysis should be based on:

(i) workflow of proposed method - you only gave a description expressed in words (lines 88-101) with reference to program code

(ii) complexity analysis of proposed algorithm/approaches

(iii) limitations that can appear in reference to input set, hardware used for test,

Your simulations are not final proofs of obtained results, but they confirm from practical side, what was achieved. Please expand this including fore theoretical framework.

Response:

Thank you for your valuable feedback, especially regarding the need for a more comprehensive analysis of the experimental results. We greatly appreciate your comments and have made significant revisions to the manuscript to provide more comprehensive analytical support for our experimental findings. The specific revisions are as follows:

We have expanded the description of the proposed method's workflow. Now, we provide a more detailed step-by-step explanation in the following sections: Workflow Optimization Speedup (lines 76-100), Parallel Compression Using pthreads (lines 101-119), and Optimized Reading with Parallel Parsing (lines 120-135). Additionally, we conducted a detailed theoretical analysis and explained the total time savings between zm3vcf and M3VCF (lines 136-137). This analysis highlights the reasons for FM3VCF's superior performance.

Furthermore, we thoroughly analyzed the overall complexity after workflow optimization and multi-threaded compression and parsing (lines 138-151). Focusing on the improvement in the compression step, we further simplified the overall complexity, particularly in the case of multi-threaded compression (lines 152-157). This simplification highlights the primary advantage of multi-threading in the compression step, where distributing the task across multiple threads significantly reduces time complexity. This leads to a substantial increase in overall execution speed, especially as the dataset size increases.

We have introduced Amdahl's Law in the manuscript to explain the theoretical limits of parallelization. This law helps explain the phenomenon where performance gains diminish as the number of threads increases, particularly when non-parallelizable parts of the task (such as I/O operations and certain overheads) begin to dominate. This theoretical framework provides a more comprehensive understanding of the factors affecting performance results (lines 248-280).

We hope that these additions address your concerns regarding the lack of analytical foundation.

Comment 2:

Line 26, there is:

FM3VCF is a powerful tool for both efficiently compressing VCF files and accelerating the loading of large VCF files in genotype data analyse.

I would prefer not to use such words as 'powerful'; we were able to see the accelaration effects but without analytical proofs that your approach will work for all input data (input cases) such strong statements are not acceptable. I'm not saying that your is wrong or does not work, but this cannot be based only on simulations - see also remark 6.

Response:

We agree with Reviewer #3’s suggestion and have adopted more cautious words. We have replaced the term "powerful" with "effective" to more accurately reflect the performance improvements observed in our experiments (lines 25-26).

Additionally, we further clarify that the expanded test datasets we used were not simply repeated multiple times but were randomly shuffled by the computer. Your feedback has been valuable, and it inspired us to further improve our experiments. We have used ped-sim(https://github.com/williamslab/ped-sim) to randomly simulate offspring genotypes based on real human data, following genetic principles, to expand the genotype dataset (lines 210-218). The simulation code is also available at https://github.com/YOUTLab/FM3VCF. This more reasonable new data showed even more significant acceleration effects in our testing scenarios (lines 219-240), and we have provided analysis and explanations for these results (lines 242-247). We have also retained the results from the previous computer-generated random dataset in Supporting Information S1 (lines 290-303), which further demonstrates the effectiveness of our software across different datasets.

Comment 3:

Technical remarks -> in introduction please separate references from words and put space between the words and literature references according to: xxx[1] -> xxx [1].

Response:

We have meticulously reviewed the manuscript to correct the citation formatting throughout. All references are now properly spaced from the preceding text, adhering to the standard citation format (e.g., "xxx [1]").

Comment 4:

Lines 150-151, we have:

This software provides both a command line interface and a C/C++ library interface,

I cannot see this in paper.

Response:

Thank you for your valuable feedback. In the Results section, we described the usage of both the command line and C/C++ library interfaces in Box 1 and Box 2. However, we realize that the original wording may not have been clear, leading to some misunderstanding. To address this, we have revised the manuscript to more clearly specify that the first and second lines of Box 2 correspond to the command line interface (lines 192-193), while lines 3 to 28 describe the C/C++ library interface (lines 195-196). We hope this revision will make the usage of both interfaces clearer.

Thank you again for your feedback. We believe these changes will make the manuscript clearer and more understandable.

Comment 5:

Is it possible to express the resuts of compressing files with the use of compressing threads in terms of mathematical formula. From Fig. 3 b) we can see after the quick increase of characteristics there is no difference bewteen 20 and 30 threads. What is the reason of such behavior? Line 131 gives information about 'sa

---

## [Decision Letter · Decision Letter 1]

25 Apr 2025

FM3VCF: A Software Library for Accelerating the Loading of Large VCF Files in Genotype Data Analyses

PONE-D-24-38236R1

Dear Dr. Tang,

We’re pleased to inform you that your manuscript has been judged scientifically suitable for publication and will be formally accepted for publication once it meets all outstanding technical requirements.

Kind regards,

Andrea Tangherloni

Academic Editor

PLOS ONE

Additional Editor Comments (optional):

Reviewers' comments:

Reviewer's Responses to Questions

**Comments to the Author**

Reviewer #1: All comments have been addressed

Reviewer #3: All comments have been addressed

2. Is the manuscript technically sound, and do the data support the conclusions?

Reviewer #1: Yes

Reviewer #3: Yes

3. Has the statistical analysis been performed appropriately and rigorously?

Reviewer #1: Yes

Reviewer #3: Yes

4. Have the authors made all data underlying the findings in their manuscript fully available?

Reviewer #1: Yes

Reviewer #3: Yes

5. Is the manuscript presented in an intelligible fashion and written in standard English?

Reviewer #1: Yes

Reviewer #3: Yes

Reviewer #1: The authors have thoroughly addressed all previously raised comments, significantly enhancing the quality and clarity of the manuscript. Their revisions have effectively resolved prior concerns, clearly reflecting careful consideration and comprehensive improvements. Specifically, the changes made to methodology, result interpretations, and overall manuscript structure have strengthened the scientific rigor and readability of the paper. The updated manuscript demonstrates meticulous attention to detail and proper integration of all suggested corrections and recommendations. Additionally, the responses provided by the authors were clear, detailed, and fully justified, showcasing their commitment to enhancing the paper's overall contribution. Therefore, given these commendable improvements, I find the manuscript suitable and recommend that it can now be accepted in its present form without requiring any further modifications.

Reviewer #3: In revised version all of my remarks were taken into account and I'm satisfied with priovided explanations.

**Do you want your identity to be public for this peer review?** For information about this choice, including consent withdrawal, please see our Privacy Policy

Reviewer #1: No

Reviewer #3: No

---

## [Editor Report · Acceptance letter]

PONE-D-24-38236R1

PLOS ONE

Dear Dr. Tang,

I'm pleased to inform you that your manuscript has been deemed suitable for publication in PLOS ONE. Congratulations! Your manuscript is now being handed over to our production team.

Kind regards,

on behalf of

Dr. Andrea Tangherloni

Academic Editor

PLOS ONE